# Prevalence of Sarcopenia Determined by Computed Tomography in Pancreatic Cancer: A Systematic Review and Meta-Analysis of Observational Studies

**DOI:** 10.3390/cancers16193356

**Published:** 2024-09-30

**Authors:** Antonio Jesús Láinez Ramos-Bossini, Antonio Gámez Martínez, David Luengo Gómez, Francisco Valverde-López, Consolación Melguizo, José Prados

**Affiliations:** 1Department of Radiology, Hospital Universitario Virgen de las Nieves, 18014 Granada, Spain; antonio.gam.mar@gmail.com (A.G.M.); davidluengog@gmail.com (D.L.G.); 2Advanced Medical Imaging Group (TeCe-22), Instituto Biosanitario de Granada, 18016 Granada, Spain; 3Department of Gastroenterology and Hepatology, Hospital Universitario Virgen de las Nieves, 18014 Granada, Spain; fcovalverde89@gmail.com; 4Department of Human Anatomy and Embryology, Faculty of Medicine, University of Granada, 18071 Granada, Spain; cmelguizo@ugr.es (C.M.); jcprados@ugr.es (J.P.); 5Institute of Biopathology and Regenerative Medicine (IBIMER), University of Granada, 18100 Granada, Spain; 6Center of Biomedical Research (CIBM), University of Granada, 18100 Granada, Spain

**Keywords:** pancreatic neoplasm, sarcopenia, computed tomography, prevalence, skeletal muscle index

## Abstract

**Simple Summary:**

Sarcopenia, a condition where muscle mass decreases, is linked to worse outcomes in pancreatic cancer patients. Computed tomography scans, which are routinely used to monitor these patients, offer a chance to assess sarcopenia without additional procedures. However, different studies report varying rates of sarcopenia due to the use of different measurement methods and thresholds. This variability can affect conclusions regarding patient outcomes, such as overall or progression-free survival. In this study, we found that sarcopenia affects nearly half of pancreatic cancer patients, with higher rates in those with advanced stages of the disease. The prevalence varied depending on the measurement methods used. These findings highlight the need to standardize how sarcopenia is assessed in future studies, which may improve clinical decision making and patient care.

**Abstract:**

**Introduction:** Sarcopenia, a condition characterized by a loss of skeletal muscle mass, is increasingly recognized as a significant factor influencing patient outcomes in pancreatic cancer (PC). This systematic review and meta-analysis aimed to estimate the prevalence of sarcopenia in patients with PC using computed tomography and to explore how different measurement methods and cut-off values impact such prevalence. **Materials and Methods:** Following the Preferred Reporting Items for Systematic Reviews and Meta-Analyses guidelines, a comprehensive search of PubMed, Web of Science, and EMBASE databases was performed, identifying 48 observational studies involving 9063 patients. **Results:** The overall pooled prevalence of sarcopenia was 45% (95% CI, 40–50%), but varied significantly by the method used: 47% when measured with the skeletal muscle index and 33% when assessed with the total psoas area. In addition, in studies using SMI, sarcopenia prevalence was 19%, 45%, and 57% for cutoff values <40 cm^2^/m^2^, 40–50 cm^2^/m^2^, and >50 cm^2^/m^2^, respectively. Moreover, the prevalence was higher in patients receiving palliative care (50%) compared to those treated with curative intent (41%). High heterogeneity was observed across all analyses, underscoring the need for standardized criteria in sarcopenia assessment. **Conclusions:** Our findings highlight the substantial variability in sarcopenia prevalence, which could influence patient outcomes, and stress the importance of consensus in measurement techniques to improve clinical decision making and research comparability.

## 1. Introduction

Pancreatic adenocarcinoma is a malignant solid tumor with a five-year survival rate of less than 10% [1]. In recent years, there has been a growing interest in the relationship between body composition and the prognosis in pancreatic cancer (PC) due to the increasing ease of obtaining this information in routine clinical practice [2]. One of the main body measures in this context is sarcopenia, a condition of decreased skeletal muscle mass that can lead to a decline in physical ability [3]. Despite the lack of worldwide agreement on the definition of sarcopenia [4], the European Working Group on Sarcopenia in the Elderly (EWGSOP) defines it as decreased muscle mass and impaired muscle function [5]. This entity, which is observed more frequently with increasing age, has been consistently associated with a poorer prognosis in various types of cancer [6], including PC [7].

Several imaging modalities, such as dual-energy X-ray absorptiometry (DXA), computed tomography (CT), magnetic resonance imaging (MRI), and ultrasound can be used to estimate muscle mass and quality, confirm the diagnosis of sarcopenia, and obtain specific measures and derived indices in a protocolized manner [8]. CT allows differentiation of body tissues by Hounsfield Units (HU) based on tissue-specific attenuation values [9]. The fact that CT is routinely performed to evaluate tumor lesions and to monitor possible distant metastases in cancer patients offers the advantage of assessing sarcopenia without the need to perform new scans or employing higher doses of ionizing radiation [4]. Currently, it is considered the gold standard method of body composition analysis and diagnosis of abnormal body composition phenotypes, especially in nutritionally vulnerable patients [10].

However, while the clinical relevance of sarcopenia in cancer management has been widely acknowledged, there remains a substantial gap in understanding how variations in its measurement and reporting can influence clinical decisions and patient outcomes. This variability is particularly significant in PC, where sarcopenia may play a critical role in treatment planning and survival prediction. In this context, different methods have been proposed to measure sarcopenia based on CT imaging. The seminal work by Prado et al. defined the Total Muscle Area (TMA) as the area of muscle determined in a slice cut at the level of L3 [11]. To control the influence of body height, the Skeletal Muscle Index (SMI) has been defined as the TMA divided by the square of the height (m^2^). This index can also be found in the literature under the acronyms of SKM [12], SKMI [13], or TSM [14]. Other measures include the Total Psoas Area [15,16,17], which is restricted to the area of the psoas at the L3 vertebral body level, the Total Psoas Volume, which corresponds to three-dimensional volumetric assessment of the psoas [18], and the Total Psoas Index, which corresponds to the TPA divided by the square of the height [19]. However, the cut-off values to define sarcopenia using each of these indices in cancer research differ among various studies due to the fact that body composition is greatly affected by different regions, ethnicities, and socio-economic factors, among other reasons [20].

The lack of standardized diagnostic criteria for sarcopenia introduces a challenge not only in research but also in clinical practice, where accurate risk stratification is critical. Inconsistent use of measurement techniques, such as differing thresholds for SMI or TPA, can lead to significant discrepancies in reported sarcopenia prevalence and its association with patient outcomes. A recent study by Wu et al. compared the prevalence of sarcopenia in patients with PC based on the criteria used by two different institutions for Western and Eastern populations [14]. They found important differences in the prevalence of sarcopenia, opening questions on the reliability of the outcomes reported by studies published to date, as they applied heterogeneous criteria which may translate into significant differences in the outcomes measured (e.g., survival, post-surgical complications), precluding comparability between studies. However, these potential differences based on the criteria selected to define sarcopenia have been scarcely explored to a large extent.

The aim of this work is to analyze the prevalence of sarcopenia based on CT-derived measurements in patients with PC and to explore the existing heterogeneity according to different variables that may introduce biases, especially the cut-off value chosen for its definition.

## 2. Methods

### 2.1. Eligibility Criteria

The selection criteria included observational studies of patients with pancreatic cancer reporting the prevalence of sarcopenia determined by CT and any survival outcomes. The Preferred Reporting Items for Systematic Reviews and Meta-Analyses (PRISMA) [21] guidelines were followed in the design and writing of the study. The protocol has not been registered. The exclusion criteria were studies reporting no mortality-related outcomes, articles published in languages other than English, studies with incomplete data on sarcopenia prevalence, editorials, letters, abstracts, or conference proceedings.

### 2.2. Information Sources and Search Strategy

Two authors (A.J.L.R.-B. and A.G.M.) searched the PubMed, Web of Science, and EMBASE databases to identify original studies published from database inception until 26 April 2024. Different search strategies were carried out and a final consistent equation was constructed as follows: “Pancreatic neoplasm[MeSH Terms] AND (“sarcopenia[MeSH Terms] OR “cachexia[MeSH Terms]” OR “body composition[MeSH Terms]”) (Appendix A). To increase the sensitivity of the search, references of all fully-read articles were also examined. No date or language restrictions were established.

All titles and abstracts of interest were screened and those which did not meet the eligibility criteria were excluded. Subsequently, the screened studies were fully read to assess whether they met all eligibility criteria. Figure 1 shows the flow diagram of the study.

### 2.3. Measured Variables and Subgroup Analyses

Data were collected regarding study characteristics (authors, year of publication, country, study design, and number of institutions involved), patient population (sample size and patients’ age and sex), sarcopenia measurement (measure index and cut-off for males and females), and cancer-related characteristics (type of PC and patient management). The primary outcome was the prevalence of sarcopenia determined by CT.

In addition, we performed subgroup analyses based on the following variables:Method used to calculate sarcopenia. In particular, we specified three subgroups: skeletal muscle index (SMI) or analogous (e.g., SKMI, SKM), and total psoas area (TPA) or analogous (e.g., TPV, PMI).For studies measured by SMI, we analyzed the differences between groups according to three intervals: <40 cm^2^/m^2^, 40–50 cm^2^/m^2^, and >50 cm^2^/m^2^. In the study by Park et al. [22], data were provided in different units (i.e., kg/m^2^) because the authors converted their data to a previously reported reference value (Appendicular Skeletal Muscle, ASM) obtained from DXA imaging, and thus, we re-calculated the reference values according to the formula used [23]: ASM/height2 (kg/m^2^) = 0.11 × SMI (cm^2^/m^2^) + 1.17. When a study reported the prevalence of sarcopenia using different cutoffs, the sample was split or duplicated accordingly and independently analyzed.Oncological context in terms of patient management, namely palliative (non-resectable or metastatic cancer) or curative (managed with surgery with or without chemo/radiotherapy).

Studies not defining any of these variables were excluded from the corresponding subgroup analysis.

### 2.4. Data Extraction

Two authors (D.LG. and F.V.L.) independently extracted the data from the selected articles and a senior author (J.P.) reviewed the data and solved any discrepancies. If there were several definitions for sarcopenia, we included the one which resulted significant for survival analyses in the study. All data were stored using a spreadsheet designed for such purpose.

### 2.5. Quality Assessment

The Newcastle Ottawa scale (NOS) [24] was applied to evaluate the quality of the included studies. The scale scores three categories with a maximum score of nine stars: selection, comparability, and outcome. An appropriate participant selection for the exposure and non-exposure cohorts is rated by four stars, the comparability of the cohort is reflected by two stars, and the evaluation of the result and follow-up is rated by three stars. Studies that scored >5 stars had moderate-to-high quality [20].

### 2.6. Statistical Analysis

We applied the DerSimonian–Laird method with a random effects model to estimate the prevalence of sarcopenia. The I^2^ statistic was used to assess heterogeneity among studies with non-relevant, moderate, and considerable cut-off values set at I^2^ < 40%, 40% < I^2^ < 75%, and I^2^ > 75%, respectively, as in previous meta-analyses [25]. Sensitivity analyses were carried out in the cases of significant heterogeneity (I^2^ > 40%) by consecutive elimination of each study in order to estimate its contribution to the pooled estimates. Publication bias was examined using funnel plotting and Egger’s test.

*p*-values <0.05 were considered statistically significant. All statistical analyses were carried out with software R (version 4.3.2 for Windows) [26] using the ‘meta’ package [27].

## 3. Results

### 3.1. Characteristics of the Included Studies

A total of 48 studies were included in the meta-analysis [12,13,14,15,16,17,18,19,22,28,29,30,31,32,33,34,35,36,37,38,39,40,41,42,43,44,45,46,47,48,49,50,51,52,53,54,55,56,57,58,59,60,61,62,63,64,65,66]. Two studies provided separated measures for their patient cohorts and were, thus, split into two different studies for analyses; the study by Nakajima et al. provided separated data for resectable and borderline resectable PC, while Wu et al. provided two independent measurements of sarcopenia based on Eastern and Western criteria [14,16]. Therefore, 50 studies were considered in data analysis, encompassing data from 9063 patients in the original cohorts (9209 patients for estimations because the sample size from the study by Wu et al. was duplicated). The percentage of women in the studies was 45%. The study with a larger sample size included 763 patients [18], and the study with the smallest sample size included 41 patients [12].

Most studies (43/50, 86%) applied SMI or an analogous measurement to estimate sarcopenia, while 7 (14%) studies applied Total Psoas Area/Volume or an analogous measurement. When SMI was the method used, 15 studies defined sarcopenia according to Prado’s criteria (i.e., males < 52.4 cm^2^/m^2^; females < 38.5 cm^2^/m^2^) [11], 7 used the Japan Society of Hepatology (JHS) criteria (<42 cm^2^/m^2^ for men and <38 cm^2^/m^2^ for women) [67], and the remainder used a different cutoff value, including lowest quartile [18,59], tertile [16,61], or median [57,58] of their samples, among others.

Information related to tumor histology and staging was not provided in detail in most studies, but many of them explicitly described PC as adenocarcinoma, and it was usually specified whether included patients were in advanced stages of the disease or not. In 26 studies, PC was managed with curative intention (surgery with or without chemo/radiotherapy), 21 studies included patients managed with palliative treatment (usually managed with palliative chemotherapy), and 3 studies did not specify the oncological management. The average quality score according to the NOS was 7.2 stars. Table 1 summarizes the main characteristics of each of the included studies in this meta-analysis.

### 3.2. Prevalence of Sarcopenia and Subgroup Analyses

#### 3.2.1. Prevalence of Sarcopenia

Sarcopenia was present in 3928 patients according to the figures provided in the included studies, implying a pooled estimate for the prevalence of sarcopenia of 42.7% (95%CI, 40–50%). The I^2^ value (95%) indicated high heterogeneity. The lowest and highest reported prevalence of sarcopenia among the included studies was 9.7% (29/299 patients) [56] and 86.4% (76/88 patients) [22], respectively. Figure 2 shows the forest plot of the pooled prevalence of sarcopenia in the study.

#### 3.2.2. Prevalence of Sarcopenia Based on the CT-Based Measurement Index Chosen in the Included Studies

In total, 7133 patients were included in the 43 studies in which sarcopenia was measured using SMI. Of these, 47% (95%CI, 42–52%) had sarcopenia. High heterogeneity was found (I^2^ = 93%). In the remaining 7 studies where sarcopenia was measured using TPA/TPV or an analogous measurement (2076 patients), the prevalence of sarcopenia was 33% (95%CI, 24–43%). High heterogeneity was also observed (I^2^ = 94%). Figure 3 shows the forest plot of this subgroup analysis.

#### 3.2.3. Prevalence of Sarcopenia Based on the Cutoff Value Chosen in the Included Studies

In the 43 studies which measured sarcopenia using SMI, three studies (7%) used cutoff values below 40 cm^2^/m^2^, and the prevalence of sarcopenia was 19% (95%CI, 5–54%). Twenty-six studies (60.5%) used cut-off values between 40 and 50 cm^2^/m^2^ and the prevalence of sarcopenia was 45% (95%CI, 40–50%). The remainder studies (32.5%) applied a cutoff value over 50 cm^2^/m^2^, with a prevalence of sarcopenia of 57% (95%CI, 51–64%). High heterogeneity was found in the three subgroups, especially in the former (I^2^ = 98%, 89% and 89%, respectively). Figure 4 shows the forest plot of this subgroup analysis.

#### 3.2.4. Prevalence of Sarcopenia Based on Treatment Intention in the Included Studies

Of the 47 studies which reported treatment intention, 26 (55.3%) included patients with PC treated with curative intention, showing a prevalence of sarcopenia of 41% (95%CI, 35–48%). The remaining studies (44.7%) included patients in a palliative setting, reporting a prevalence of sarcopenia of 50% (95%CI, 43–57%). High heterogeneity was found in both groups (I^2^ = 96% and 92%, respectively). Figure 5 shows the forest plot of this subgroup analysis.

### 3.3. Sensitivity Analysis and Publication Bias

The overall prevalence estimate of sarcopenia remained consistent, with only minor variations in the pooled prevalence when individual studies were excluded. The random effects model yielded a pooled prevalence of sarcopenia ranging from 44.43% to 46.67% across the different iterations. The heterogeneity remained high (I^2^ > 94%) throughout the analyses, indicating substantial variation between studies. No single study was found to disproportionately influence the overall meta-analysis results. Data regarding the sensitivity analysis can be found in Appendix A.

The funnel plot revealed that several studies were located outside the expected triangular region, indicating high heterogeneity among the included studies rather than publication bias. The Egger’s test showed a non-significant result (z = 0.578, *p* = 0.563), suggesting no strong statistical evidence of funnel plot asymmetry and unlikeliness of publication bias. Figure 6 shows the funnel plot of the study.

## 4. Discussion

This meta-analysis analyzed differences in CT-based sarcopenia prevalence across 48 published studies encompassing data from more than 9000 patients. The pooled prevalence of sarcopenia was 45% (95%CI, 40–50%). When SMI was the method used, the prevalence of sarcopenia was 47%, while when TPA or a similar method was used, the prevalence of sarcopenia decreased to 33%. We also explored the influence of the cutoff value selected in the studies that applied SMI, since this was the most frequently used method. We found relevant differences in sarcopenia prevalence, ranging from 19% (cutoff for males below 40 cm^2^/m^2^) to 57% (cutoff for males below 50 cm^2^/m^2^), with intermediate values (45%, 95% CI, 41–51%) for cutoff for males between 40 and 50 cm^2^/m^2^. Finally, to explore the influence of the oncological status of patients, we also analyzed the prevalence of sarcopenia in patients managed with curative and palliative intention, with values of 41% and 50%, respectively. Notably, we found a high heterogeneity (I^2^ > 85%) in all the analyses performed, which was not significantly modified in the sensitivity analysis. Our results emphasize the ample variability in the measurements of sarcopenia based on CT, highlighting the need for more consistent study designs and consensus on cutoff values.

Despite several studies have consistently reported an association between preoperative sarcopenia and PC survival outcomes [20], currently, there is no universal consensus on the gold standard method to define sarcopenia based on CT imaging. Similar to the findings of previous meta-analyses [68], we found that the most consistent CT-based method used is the SMI, which consists of measuring the cross-sectional area of all skeletal muscles at the level of the L3 vertebral body, divided by the square of height [69]. Prado et al. were the first to establish relatively widely accepted cut-off values to define sarcopenia (i.e., below 52.4 cm^2^/m^2^ and 38.5 cm^2^/m^2^ for men and women, respectively) and found that it was associated with increased mortality for solid tumors in a large population-based study [11]. However, other authors and societies have established different cutoffs, being as low as 36.2 cm^2^/m^2^ [70]. This has resulted in a wide variability of sarcopenia prevalence among studies, ranging from approximately 17% to 63% [71].

Other relatively common measures in this setting are limited to the cross-sectional area or volume of the psoas (e.g., TPA, TPV, and PMI), and variability also exists regarding the optimal cutoff value for these measurements across studies. Of note, determining the cross-sectional area can be performed through manual, semiautomatic, or fully automatic segmentation, which may introduce uncontrolled bias between studies, particularly in the absence of expert supervision, as highlighted by Tagliafico et al. [8]. Although most studies ensured an expert overview of segmentation, some studies did not report specific information on this fact. For instance, Amini et al. did not indicate if there was expert supervision in their semi-automatic image analysis [18], and Basile et al. [55] not only did not report this information, but none of the authors were affiliated with a radiology department.

On the other hand, we found that the prevalence of sarcopenia can vary substantially based on the method used (33% in studies using psoas-related measurements vs. 47% in SMI-related measurements). However, the high heterogeneity among studies needs to be taken into account. For instance, within the studies using SMI, Cho et al. [56] found a very low (10%) prevalence of sarcopenia, as compared to the high (86%) prevalence found by Park et al. [22]. A meta-analysis by Griffin et al. [2], including more than 5000 patients with resectable and borderline resectable PC found a pooled prevalence of sarcopenia of 39% (range, 14–74%), which was slightly higher than a wider meta-analysis involving a number of solid tumors performed by Surov et al., who found a 35.3% prevalence of sarcopenia [72]. Notably, the authors reported high heterogeneity (I^2^ = 93%) that did not improve significantly when controlling for assessment method, including SMI and psoas index. As they included patients with resectable PC, their results can be extrapolated to our subgroup of patients managed with curative intention, and thus, the prevalence they found is in agreement with our findings, i.e., 41% prevalence of sarcopenia in patients treated with curative intention.

Interestingly, we found some critical confusions in previous works regarding the method and cutoff used to define sarcopenia. For instance, Hou et al. [17] applied TPI (TPA divided by the square of the height), but they used the cutoff value provided by Prado et al. to define sarcopenia, which was obtained for SMI (i.e., including the area of the psoas, erector spinae, quadratus lumborum, transversus abdominis, external and internal oblique muscles of the abdomen, and rectus abdominis muscles) [11,55].

Similarly, the cutoff value used to define sarcopenia is also essential to determine its prevalence. In spite of the fact that international societies such as the EWGSOP or the JHS have tried to set standard values in this context, we found substantial variability in the definition of the established cutoff. In our meta-analysis, 15 studies used the criteria proposed by Prado et al., while 7 studies followed the threshold proposed by the JHS. Several authors opted to establish their own threshold based on their cohort data, grounded on the lack of universal consensus along with the variability dependent on factors such as ethnicity and regions [66]. Unfortunately, the criteria to establish such thresholds are also greatly variable. For instance, Peng et al. (2021) [59] and Rom et al. [60] selected a cutoff value based on the lowest interquartile range of their sample, Van Dijk et al. [66] and Choi et al. [61] based their cutoff on the lowest tertile values of their cohort, while Masuda [58] and Nowak et al. [57] used the median value. Other authors, such as Okumura et al. (2017) [62], Shimura et al. [64], or Özkul et al. [65] applied optimal stratification methods based on area under the curve values. This resulted in significant disparities that complicate even more the comparability of studies, calling for the need of standardization.

To our knowledge, only Wu et al. [14] specifically analyzed the differences in sarcopenia prevalence based on two different criteria that had been proposed by international societies in this setting—namely Western, i.e., Prado et al. [11], and Eastern, i.e., Fujiwara et al. [70]. They applied both criteria to their cohort of 146 patients and found obvious differences in sarcopenia prevalence (66.4% vs. 11%, respectively). In an effort to illustrate the influence and variability of this measure, we analyzed the prevalence of sarcopenia stratifying by cutoff value intervals below 40 cm^2^/m^2^, between 40 and 50 cm^2^/m^2^, and below 50 cm^2^/m^2^. As expected, the prevalence of sarcopenia increased from 19% to 57%. These differences are not as exaggerated as the ones found by Wu et al. [14], but they still represent obvious differences due to the influence of the cutoff value used. These results highlight the fact that selecting a method (e.g., SMI) as the standard measurement for sarcopenia does not suffice; standardizing the exact threshold should also be mandatory. Considering the epidemiological differences among populations, different criteria could be applied to Western or Eastern criteria, but this can limit comparability.

Regarding the influence of the oncological setting of patients, previous studies have emphasized the fact that patients with resectable PC, usually managed with curative intention, show lower rates of sarcopenia compared to patients with more advanced stages of the disease (i.e., usually managed with palliative intention). This seems rather logical due to the deleterious effect of cancer in body composition. A previous meta-analysis including 23 studies found that the prevalence of sarcopenia in the curative setting ranged from 17.2% to 58.7%, while in the palliative setting the range was from 9.7% to 87.8% [68]. Our meta-analysis found a 9% higher prevalence of sarcopenia in patients with PC in a palliative setting compared to the curative setting, with ranges from 35–48% to 43–57%, respectively. However, it should be noted that a high heterogeneity was also observed within each group, indicating that other factors also may play a role in this prevalence. In fact, apart from sex and BMI (i.e., height and weight), there is limited information on the influence of other factors that could play a role in the prevalence of sarcopenia, as is the case with tumor histology, ECOG, and racial and socio-economic factors. Similarly, the role of pre- and post-treatment sarcopenia should also be explored, as patients might serve as their own control, which reduces the variability due to sarcopenia cutoff values based on heterogeneous populations. To date, a very limited number of studies have focused on this comparison, as is the case with Choi et al. (2015) [30].

The main strengths of this study lie in its focus on the prevalence of sarcopenia regardless of the clinical outcome analyzed in the studies, the stratification analysis based on intervals for the cutoff values which facilitates comparability across studies, and the high number of studies included, being the largest meta-analysis on the prevalence of sarcopenia to date. The main limitations of this study lie in the retrospective nature of the included studies (all but one were retrospective and performed in a single institution), the high heterogeneity found, the establishment of cutoff ranges for SMI based on a subjective decision (mainly guided by easiness of interpretability), and the fact that only one potential confounder (i.e., oncological status) was analyzed. Other potentially relevant variables, such as specific tumor histology or stage, were not addressed due to the limited information provided in the original publications. Future studies should provide more information on these variables to allow for a more fine-tuned analysis. Apart from addressing current limitations, future studies should further explore the role of other advanced image-based analysis techniques that could improve sarcopenia-related risk stratification in cancer patients, as is the case with radiomics [73].

## 5. Conclusions

There is high variability in the methods and cutoff values for measuring sarcopenia by CT. This results in high heterogeneity among studies and significant variability in the prevalence of sarcopenia, which precludes appropriate comparability between studies. The influence of other factors, as is the case with the oncological status of patients, is also remarkable and needs to be quantified and homogenize. Our results highlight the need for reaching international consensus on the method, thresholds, and identification of confounders to appropriately analyze the influence of sarcopenia in pancreatic cancer. These limitations should also be explored in other oncological conditions and scenarios, given the increasing research interest in determining the role of sarcopenia in cancer.

## Figures and Tables

**Figure 1 cancers-16-03356-f001:**
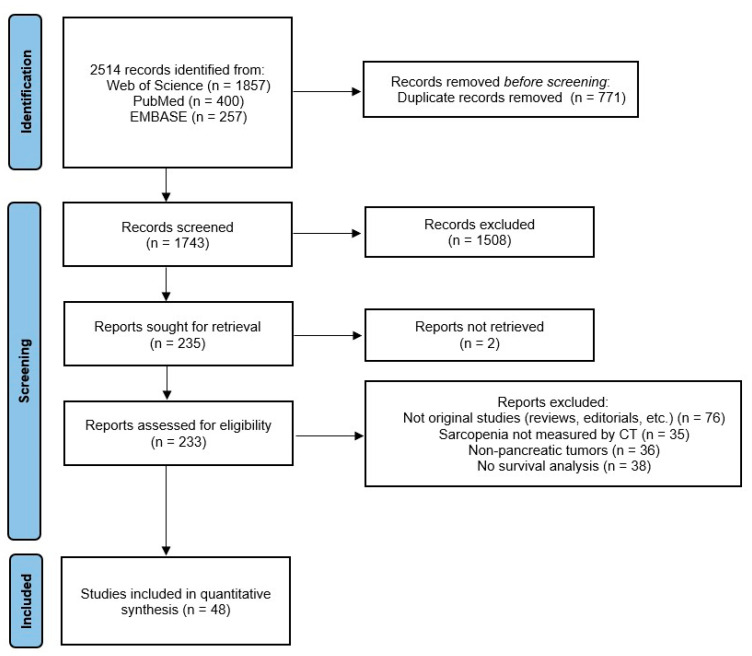
Flow diagram of the study according to the PRISMA guidelines [21]. From an initial identification of 2514 records (1743 non-duplicated articles), the screening process led to the inclusion of 48 original studies in the meta-analysis.

**Figure 2 cancers-16-03356-f002:**
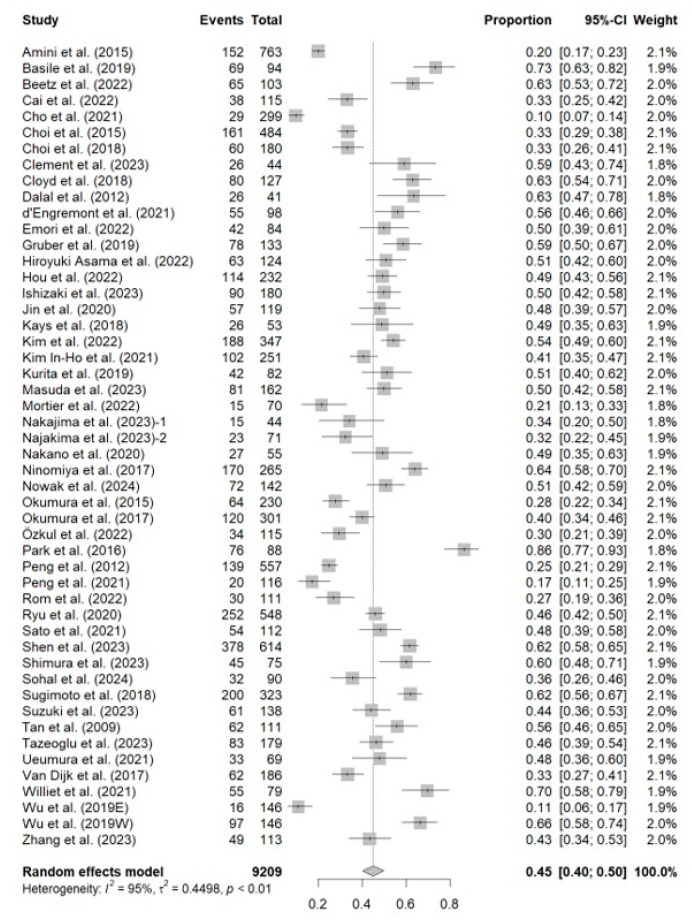
Forest plot of the pooled prevalence of sarcopenia across all studies included in the meta-analysis. Events, total number of patients with sarcopenia in each study. Total, total sample size analyzed in the study. Proportion, Events/Total. 95%-CI, 95% confidence interval for the proportion. Weight, relative weight of the study in the pooled analysis. Amini et al. (2015) [18]; Basile et al. (2019) [55]; Beetz et al. (2022) [28]; Cai et al. (2022) [29]; Cho et al. (2021) [56]; Choi et al. (2015) [30]; Choi et al. (2018) [61]; Clement et al. (2023) [31]; Cloyd et al. (2018) [32]; Dalal et al. (2012) [12]; d‘Engremont et al. (2021) [33]; Emori et al. (2022) [34]; Gruber et al. (2019) [35]; Hiroyuki Asama et al. (2022) [36]; Hou et al. (2022) [17]; Ishizaki et al. (2023) [37]; Jin et al. (2020) [38]; Kays et al. (2018) [13]; Kim et al. (2022) [39]; Kim In-Ho et al. (2021) [40]; Kurita et al. (2019) [41]; Masuda et al. (2023) [58]; Mortier et al. (2022) [42]; Nakajima et al. (2023)-1 (resectable cancer) [16]; Najakima et al. (2023)-2 (borderline resectable cancer) [16]; Nakano et al. (2020) [43]; Ninomiya et al. (2017) [44]; Nowak et al. (2024) [57]; Okumura et al. (2015) [63]; Okumura et al. (2017) [62]; Özkul et al. (2022) [65]; Park et al. (2016) [22]; Peng et al. (2012) [15]; Peng et al. (2021) [59]; Rom et al. (2022) [60]; Ryu et al. (2020) [45]; Sato et al. (2021) [46]; Shen et al. (2023) [47]; Shimura et al. (2023) [64]; Sohal et al. (2024) [48]; Sugimoto et al. (2018) [49]; Suzuki et al. (2023) [50]; Tan et al. (2009) [51]; Tazeoglu et al. (2023) [52]; Uemura et al. (2021) [53]; Van Dijk et al. (2017) [66]; Williet et al. (2021) [19]; Wu et al. (2019E [eastern criteria]) [14]; Wu et al. (2019W [western criteria]) [14]; Zhang et al. (2023) [54].

**Figure 3 cancers-16-03356-f003:**
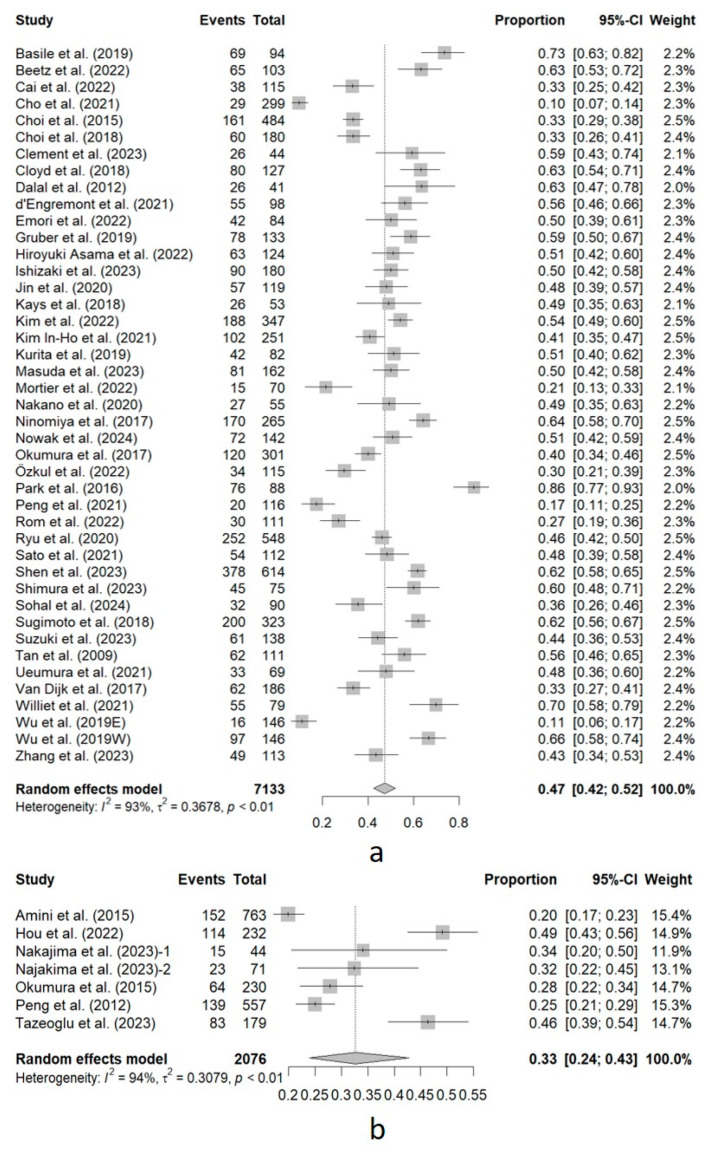
Forest plot of the subgroup analysis of sarcopenia prevalence based on the CT-based measurement index. (**a**) Studies using the skeletal muscle index or an analogous measurement to define sarcopenia. (**b**) Studies using total psoas area/volume or an analogous measurement to define sarcopenia. Events, total number of patients with sarcopenia in each study. Total, total sample size analyzed in the study. Proportion, Events/Total. 95%-CI, 95% confidence interval for the proportion. Weight, relative weight of the study in the pooled analysis. Amini et al. (2015) [18]; Basile et al. (2019) [55]; Beetz et al. (2022) [28]; Cai et al. (2022) [29]; Cho et al. (2021) [56]; Choi et al. (2015) [30]; Choi et al. (2018) [61]; Clement et al. (2023) [31]; Cloyd et al. (2018) [32]; Dalal et al. (2012) [12]; d‘Engremont et al. (2021) [33]; Emori et al. (2022) [34]; Gruber et al. (2019) [35]; Hiroyuki Asama et al. (2022) [36]; Hou et al. (2022) [17]; Ishizaki et al. (2023) [37]; Jin et al. (2020) [38]; Kays et al. (2018) [13]; Kim et al. (2022) [39]; Kim In-Ho et al. (2021) [40]; Kurita et al. (2019) [41]; Masuda et al. (2023) [58]; Mortier et al. (2022) [42]; Nakajima et al. (2023)-1 (resectable cancer) [16]; Najakima et al. (2023)-2 (borderline resectable cancer) [16]; Nakano et al. (2020) [43]; Ninomiya et al. (2017) [44]; Nowak et al. (2024) [57]; Okumura et al. (2015) [63]; Okumura et al. (2017) [62]; Özkul et al. (2022) [65]; Park et al. (2016) [22]; Peng et al. (2012) [15]; Peng et al. (2021) [59]; Rom et al. (2022) [60]; Ryu et al. (2020) [45]; Sato et al. (2021) [46]; Shen et al. (2023) [47]; Shimura et al. (2023) [64]; Sohal et al. (2024) [48]; Sugimoto et al. (2018) [49]; Suzuki et al. (2023) [50]; Tan et al. (2009) [51]; Tazeoglu et al. (2023) [52]; Uemura et al. (2021) [53]; Van Dijk et al. (2017) [66]; Williet et al. (2021) [19]; Wu et al. (2019E [eastern criteria]) [14]; Wu et al. (2019W [western criteria]) [14]; Zhang et al. (2023) [54].

**Figure 4 cancers-16-03356-f004:**
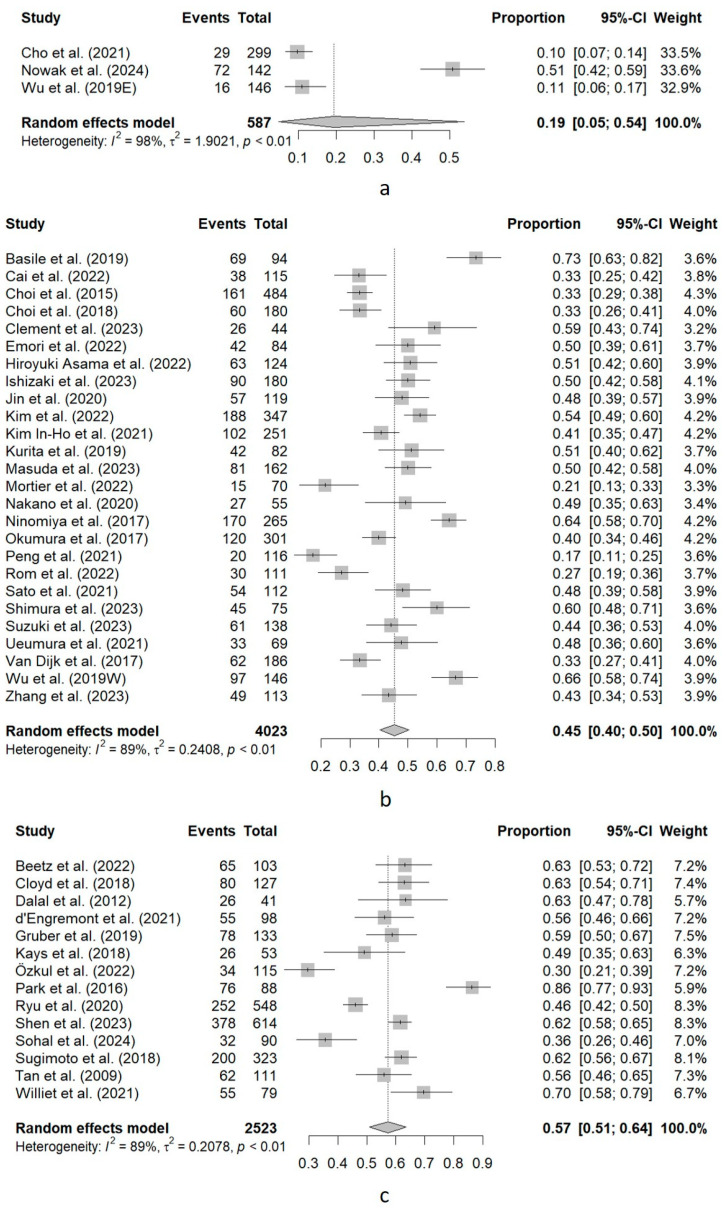
Forest plot of the subgroup analysis of sarcopenia prevalence based on the cutoff value for males used in each study (skeletal muscle index, SMI). (**a**) Cutoff value for SMI <40 cm^2^/m^2^. (**b**) Cutoff value for SMI between 40 and 50 cm^2^/m^2^. (**c**) Cutoff value for SMI over 50 cm^2^/m^2^. Events, total number of patients with sarcopenia in each study. Total, total sample size analyzed in the study. Proportion, Events/Total. 95%-CI, 95% confidence interval for the proportion. Weight, relative weight of the study in the pooled analysis. Basile et al. (2019) [55]; Beetz et al. (2022) [28]; Cai et al. (2022) [29]; Cho et al. (2021) [56]; Choi et al. (2015) [30]; Choi et al. (2018) [61]; Clement et al. (2023) [31]; Cloyd et al. (2018) [32]; Dalal et al. (2012) [12]; d‘Engremont et al. (2021) [33]; Emori et al. (2022) [34]; Gruber et al. (2019) [35]; Hiroyuki Asama et al. (2022) [36]; Ishizaki et al. (2023) [37]; Jin et al. (2020) [38]; Kays et al. (2018) [13]; Kim et al. (2022) [39]; Kim In-Ho et al. (2021) [40]; Kurita et al. (2019) [41]; Masuda et al. (2023) [58]; Mortier et al. (2022) [42]; Nakano et al. (2020) [43]; Ninomiya et al. (2017) [44]; Nowak et al. (2024) [57]; Okumura et al. (2017) [62]; Özkul et al. (2022) [65]; Park et al. (2016) [22]; Peng et al. (2021) [59]; Rom et al. (2022) [60]; Ryu et al. (2020) [45]; Sato et al. (2021) [46]; Shen et al. (2023) [47]; Shimura et al. (2023) [64]; Sohal et al. (2024) [48]; Sugimoto et al. (2018) [49]; Suzuki et al. (2023) [50]; Tan et al. (2009) [51]; Uemura et al. (2021) [53]; Van Dijk et al. (2017) [66]; Williet et al. (2021) [19]; Wu et al. (2019E [eastern criteria]) [14]; Wu et al. (2019W [western criteria]) [14]; Zhang et al. (2023) [54].

**Figure 5 cancers-16-03356-f005:**
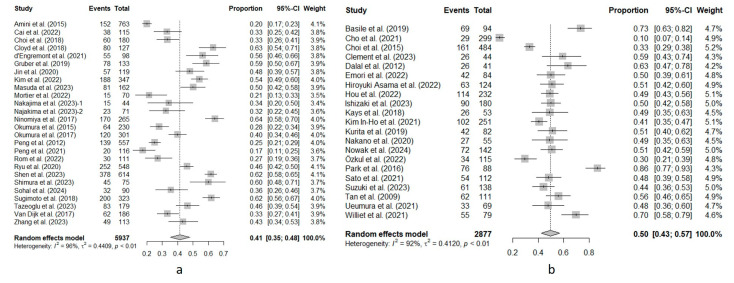
Forest plot of the subgroup analysis of sarcopenia prevalence based on the oncological status of patients. (**a**) Prevalence of sarcopenia in patients with pancreatic cancer managed with curative intention. (**b**) Prevalence of sarcopenia in patients with pancreatic cancer managed with palliative treatment. Events, total number of patients with sarcopenia in each study. Total, total sample size analyzed in the study. Proportion, Events/Total. 95%-CI, 95% confidence interval for the proportion. Weight, relative weight of the study in the pooled analysis. Amini et al. (2015) [18]; Basile et al. (2019) [55]; Cai et al. (2022) [29]; Cho et al. (2021) [56]; Choi et al. (2015) [30]; Choi et al. (2018) [61]; Clement et al. (2023) [31]; Cloyd et al. (2018) [32]; Dalal et al. (2012) [12]; d‘Engremont et al. (2021) [33]; Emori et al. (2022) [34]; Gruber et al. (2019) [35]; Hiroyuki Asama et al. (2022) [36]; Hou et al. (2022) [17]; Ishizaki et al. (2023) [37]; Jin et al. (2020) [38]; Kays et al. (2018) [13]; Kim et al. (2022) [39]; Kim In-Ho et al. (2021) [40]; Kurita et al. (2019) [41]; Masuda et al. (2023) [58]; Mortier et al. (2022) [42]; Nakajima et al. (2023)-1 (resectable cancer) [16]; Najakima et al. (2023)-2 (borderline resectable cancer) [16]; Nakano et al. (2020) [43]; Ninomiya et al. (2017) [44]; Nowak et al. (2024) [57]; Okumura et al. (2015) [63]; Okumura et al. (2017) [62]; Özkul et al. (2022) [65]; Park et al. (2016) [22]; Peng et al. (2012) [15]; Peng et al. (2021) [59]; Rom et al. (2022) [60]; Ryu et al. (2020) [45]; Sato et al. (2021) [46]; Shen et al. (2023) [47]; Shimura et al. (2023) [64]; Sohal et al. (2024) [48]; Sugimoto et al. (2018) [49]; Suzuki et al. (2023) [50]; Tan et al. (2009) [51]; Tazeoglu et al. (2023) [52]; Uemura et al. (2021) [53]; Van Dijk et al. (2017) [66]; Williet et al. (2021) [19]; Zhang et al. (2023) [54].

**Figure 6 cancers-16-03356-f006:**
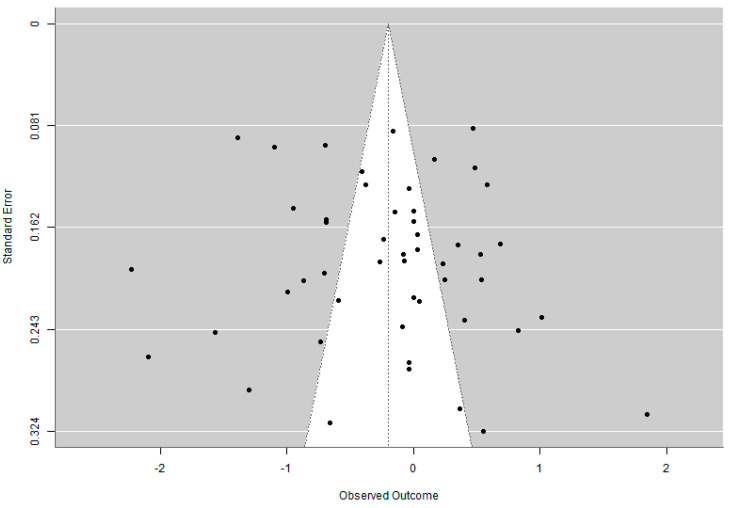
Funnel plot to visually assess the presence of publication bias in the studies included in the meta-analysis. Each dot represents a single study, plotted according to its observed outcome and standard error. The presence of several studies outside the expected triangular region indicates high heterogeneity among the included studies rather than publication bias.

**Table 1 cancers-16-03356-t001:** Main characteristics of the studies included in this meta-analysis. ASM, Appendicular Skeletal Muscle. F, female. M, male. m (IQR), median (interquartile range). NOS, Newcastle–Ottawa Scale. PC, pancreatic cancer. PDAC, pancreatic ductal adenocarcinoma. PMI, psoas muscle mass index. SMI, skeletal muscle mass index. TPV, total psoas volume. TSM, total skeletal muscle index. X ± SD, mean ± standard deviation. * From data calculation provided in the methodology of the article, the corresponding values for class I sarcopenia are 57.5 cm^2^/m^2^ and 38.3 cm^2^/m^2^ for men and women, respectively.

Author (Year)	Country	Design	Number of Institutions	N	Age m (IQR) X ± SD	Women (%)	Sarcopenia (n)	Sarcopenia (%)	Imaging Index for Sarcopenia	Definition of Cut-Off Value	Sex-Specific Cut-Off Values	Tumor Information	Management	NOS
Amini et al. (2015) [18]	USA	Retrospective	Single-center	763	67 (58–74)	45.2	152	19.9	TPV	Lowest quartile	M < 17.2 cm^2^/m^2^ F < 12.0 cm^2^/m^2^	PDAC	Curative	6★
Basile et al. (2019) [55]	Italy	Retrospective	Single-center	94	45 (48% <70 years)	44.6	69	73.4	SMI	Prado et al.	M < 43 cm^2^/m^2^ (BMI < 25); 53 cm^2^/m^2^ (BMI > 25) F < 41 cm^2^/m^2^	Advanced PC	Palliative	7★
Beetz et al. (2022) [28]	Germany	Retrospective	Single-center	103	62 + 11 (37–84)	39.8	65	63.1	SMI	Prado et al.	M < 52.3 cm^2^/m^2^ F < 38.5 cm^2^/m^2^	PDAC	Not specified	7★
Cai et al. (2022) [29]	China	Retrospective	Single-center	115	65.1 + 9	38.2	38	33	SMI	AUC (best accuracy, outcome: ‘mortality’)	M < 45.16 cm^2^/m^2^ F <34.65 cm^2^/m^2^	PDAC	Curative	8★
Cho et al. (2021) [56]	Korea	Retrospective	Single-center	299	62 (35–83)	40.4	29	9.6	SMI	Fujiwara et al.	M < 36.2 cm^2^/m^2^ F < 29.6 cm^2^/m^2^	Locally advanced PC	Palliative	8★
Choi et al. (2015) [30]	Korea	Retrospective	Single-center	484	60.4 (20–85)	39	161	33.2	SMI	AUC (not specified)	M < 42.2 cm^2^/m^2^ F < 33.9 cm^2^/m^2^	Advanced PC	Palliative	8★
Choi et al. (2018) [61]	Korea	Retrospective	Single-center	180	64.4 + 9.3	45.5	60	33.3	SMI	Lowest tertile	M < 45.3 cm^2^/m^2^ F < 39.3 cm^2^/m^2^	PC	Curative	7★
Clement et al. (2023) [31]	UK	Retrospective	Single-center	44	62 (52–68)	52	26	59	SMI	Prado et al.	M < 43 cm^2^/m^2^ (BMI < 25); <53 (BMI > 25) F < 41 cm^2^/m^2^	Metastatic PC	Palliative	8★
Cloyd et al. (2018) [32]	USA	Retrospective	Single-center	127	64.6 + 8.9	59	80	62.9	SKM (=SMI)	Mourtzakis et al.	M < 38.9 cm^2^/m^2^ F < 55.4 cm^2^/m^2^	PDAC	Curative	7★
Dalal et al. (2012) [12]	USA	Retrospective	Single-center	41	59 (42–81)	56	26	63.4	SKM (=SMI)	Prado et al.	M < 52.4 cm^2^/m^2^ F < 38.5 cm^2^/m^2^	Locally advanced PC	Palliative	6★
d‘Engremont et al. (2021) [33]	France	Retrospective	Single-center	98	67.7 (61.8–73.8)	47.8	55	56.1	SMI	Prado et al.	M < 52.4 cm^2^/m^2^ F < 38.5 cm^2^/m^2^	Localized PDAC	Curative	7★
Emori et al. (2022) [34]	Japan	Retrospective	Single-center	84	<65: 30 (36%) >65: 54 (64%)	36.9	42	50	SMI	Nishikawa et al.	M < 42 cm^2^/m^2^ F < 38 cm^2^/m^2^	Unresectable PDAC	Palliative	7★
Gruber et al. (2019) [35]	Austria	Retrospective	Single-center	133	68 (34–87)	48.8	78	58.6	SMI	Prado et al.	M < 52.4 cm^2^/m^2^ F < 38.5 cm^2^/m^2^	PDAC	Curative	6★
Hiroyuki Asama et al. (2022) [36]	Japan	Retrospective	Single-center	124	69 (40–84)	45.9	63	50.8	SMI	Nishikawa et al.	M < 42 cm^2^/m^2^ F < 38 cm^2^/m^2^	Unresectable PDAC	Palliative	8★
Hou et al. (2022) [17]	Taiwan	Retrospective	Single-center	232	<65: 139 (59.9) >65 = 93 (40.1)	35.7	114	49.1	TPA	Prado et al.	M < 545 mm^2^/m^2^ F < 385 mm^2^/m^2^	Advanced PC	Palliative	7★
Ishizaki et al. (2023) [37]	Japan	Retrospective	Single-center	180	<65: 90 (50%) >65: 90 (50%)	43.8	90	50	SMI	Nishikawa et al.	M < 42 cm^2^/m^2^ F < 38 cm^2^/m^2^	Unresectable PC	Palliative	8★
Jin et al. (2020) [38]	China	Retrospective	Single-center	119	60.2 + 8.4	50.4	57	47.8	SMI	Nishikawa et al.	M < 41 cm^2^/m^2^ F < 38.5 cm^2^/m^2^	Potentially resectable PDAC	Curative	7★
Kays et al. (2018) [13]	USA	Retrospective	Single-center	53	59.5 + 9.9	37.7	26	49	SKMI (=SMI)	Prado et al.	M < 52.4 cm^2^/m^2^ F < 38.5 cm^2^/m^2^	Advanced PC	Palliative	6★
Kim et al. (2022) [39]	Korea	Retrospective	Single-center	347	63.6 + 9.6	41.7	188	54.1	SMI	Prado et al.	M < 42.2 cm^2^/m^2^ F < 33.9 cm^2^/m^2^	PDAC	Curative	7★
Kim In-Ho et al. (2021) [40]	Korea	Retrospective	Single-center	251	63.4 + 9.4	35.8	102	40.6	SMI	outcome-based Contal and O‘Quigley method	M < 43 cm^2^/m^2^ (BMI < 25); <53 (BMI > 25) F < 41 cm^2^/m^2^	Metastatic PC	Palliative	6★
Kurita et al. (2019) [41]	Japan	Retrospective	Single-center	82	64 (40–80)	26.8	42	51.2	SMI	Optimum stratification (log-rank, outcome: ‘mortality’)	M < 45.3 cm^2^/m^2^ F < 37.1 cm^2^/m^2^	PC	Palliative	7★
Masuda et al. (2023) [58]	Japan	Retrospective	Single-center	162	69 (40–85)	44.4	81	50	SMI	Median value	M < 41.9 cm^2^/m^2^ F < 36.6 cm^2^/m^2^	Localized PDAC	Curative	8★
Mortier et al. (2022) [42]	France	Retrospective	Single-center	70	Sarcopenic: 65 (43–85) Non-sarcopenic: 73 (54–80)	52.8	15	21.4	SMI	Prado et al.	M < 43 cm^2^/m^2^ (BMI < 25); <53 (BMI > 25) F < 41 cm^2^/m^2^	Localized PDAC	Curative	8★
Nakajima et al. (2023)-1 [16]	Japan	Retrospective	Single-center	44	72 (65–76)	61.3	15	34	TPA	Lowest tertile	M < 7.79 cm^2^/m^2^ F < 5.70 cm^2^/m^2^	Resectable PC	Curative	7★
Najakima et al. (2023)-2 [16]	Japan	Retrospective	Single-center	71	67 (60–72)	59.1	23	32.3	TPA	Lowest tertile	M < 7.16 cm^2^/m^2^ F < 6.44 cm^2^/m^2^	Borderline resectable PC	Curative	7★
Nakano et al. (2020) [43]	Japan	Retrospective	Single-center	55	67 (35–85)	23.6	27	49	SMI	Choi et al.	M < 42.2 cm^2^/m^2^ F < 33.9 cm^2^/m^2^	Advanced PC	Palliative	8★
Ninomiya et al. (2017) [44]	Japan	Retrospective	Single-center	265	65.4 + 10.1	38.1	170	64.1	SMI	Prado et al.	M < 43.75 cm^2^/m^2^ F < 38.5 cm^2^/m^2^	PDAC	Curative	7★
Nowak et al. (2024) [57]	Germany	Retrospective	Single-center	142	64.1 + 10.5	51.4	72	50.7	SMI	Median value	M < 13.5 cm^2^/m^2^ F < 11.7 cm^2^/m^2^	Advanced PC	Palliative	8★
Okumura et al. (2015) [63]	Japan	Retrospective	Single-center	230	67 (32–87)	46	64	27.8	PMI	AUC (best accuracy, outcome: ‘death’)	M < 5.9 cm^2^/m^2^ F < 4.1 cm^2^/m^2^	PDAC	Curative	7★
Okumura et al. (2017) [62]	Japan	Retrospective	Single-center	301	68 (61–74)	44.1	120	39.8	SMI	AUC (best accuracy, outcome ‘death’)	M < 47.1 cm^2^/m^2^ F < 36.6 cm^2^/m^2^	PC	Curative	7★
Özkul et al. (2022) [65]	Turkey	Retrospective	Single-center	115	65.5 + 10.3	29.5	34	29.5	SMI	AUC (best accuracy, outcome: `mortality`)	M < 56.44 cm^2^/m^2^ F < 43.56 cm^2^/m^2^	Unresectable PC	Palliative	8★
Park et al. (2016) [22]	Korea	Retrospective	Single-center	88	65 (34–83)	32.9	76	86.3	ASM (=SMI)	Conversion from SMI to ASM. <1 SD for young adults	M < 7.50 kg/m^2^; F < 5.38 kg/m^2^ (sarcopenia class I *)	PC	Palliative	7★
Peng et al. (2012) [15]	China	Retrospective	Single-center	557	65.7 + 10.6	46.8	139	24.9	TPA	Choi et al.	M < 4.92 cm^2^/m^2^ F < 3.62 cm^2^/m^2^	PC	Curative	6★
Peng et al. (2021) [59]	China	Retrospective	Single-center	116	66.2 + 11.9	41.3	20	17.2	SMI	Lowest quartile	M < 42.2 cm^2^/m^2^ F < 33.9 cm^2^/m^2^	PC	Curative	7★
Rom et al. (2022) [60]	Israel	Retrospective	Single-center	111	67 (61–75)	46.8	30	27	SMI	Lowest quartile	M < 44.35 cm^2^/m^2^ F < 34.82 cm^2^/m^2^	PDAC	Curative	7★
Ryu et al. (2020) [45]	Korea	Retrospective	Single-center	548	62.51 (24–88)	40.5	252	45.9	SMI	Moon et al.	M < 50.18 cm^2^/m^2^ F < 38.63 cm^2^/m^2^	PC (head of pancreas)	Curative	7★
Sato et al. (2021) [46]	Japan	Retrospective	Single-center	112	67.7 (59.2–72.3)	51.7	54	48.2	SMI	Nishikawa et al.	M < 42 cm^2^/m^2^ F < 38 cm^2^/m^2^	Advanced PDAC	Palliative	7★
Shen et al. (2023) [47]	China	Retrospective	Single-center	614	59.9 + 10.3	40	378	61.5	SMI	Prado et al.	M < 52.4 cm^2^/m^2^ F < 38.5 cm^2^/m^2^	PDAC	Curative	8★
Shimura et al. (2023) [64]	Japan	Retrospective	Single-center	75	67 + 7.8	46.6	45	60	SMI	AUC	M < 48.4 cm^2^/m^2^ F < 35.5 cm^2^/m^2^	PC	Curative	8★
Sohal et al. (2024) [48]	USA	Prospective	Multi-center	90	63.2 + 8.5	54.4	32	35.5	SMI (SMA/BMI)	Not specified (=Prado et al.)	M < 52 cm^2^/m^2^ F < 39 cm^2^/m^2^	Resectable PDAC	Curative	8★
Sugimoto et al. (2018) [49]	USA	Retrospective	Single-center	323	65 (38–88)	45.5	200	61.9	SMI	Fearon et al. (=Prado et al.)	M < 55.4 cm^2^/m^2^ F < 38.9 cm^2^/m^2^	PDAC	Curative	7★
Suzuki et al. (2023) [50]	Japan	Retrospective	Single-center	138	67.5 (59.7–74)	42	61	44.2	SMI	Nishikawa et al.	M < 42 cm^2^/m^2^ F < 38 cm^2^/m^2^	Unresectable PC	Palliative	8★
Tan et al. (2009) [51]	Canada	Retrospective	Single-center	111	64.4 + 9.3	53.1	62	55.8	SMI	Prado et al.	M < 59.1 cm^2^/m^2^ F < 48.4 cm^2^/m^2^	PC	Palliative	6★
Tazeoglu et al. (2023) [52]	Turkey	Retrospective	Single-center	179	60.45 + 13.08	41.3	83	46.3	PMI	Bahat et al.	M < 5.3 cm^2^/m^2^ F < 3.6 cm^2^/m^2^	PC	Curative	8★
Uemura et al. (2021) [53]	Japan	Retrospective	Single-center	69	63 (38–74)	44.9	33	47.8	SMI	Nishikawa et al.	M < 42 cm^2^/m^2^ F < 38 cm^2^/m^2^	Advanced PC	Palliative	7★
Van Dijk et al. (2017) [66]	The Netherlands	Retrospective	Single-center	186	66.5	45.1	62	33.3	L3-muscle attenuation index (=SMI)	Lowest tertile	M < 45.1 cm^2^/m^2^ F < 36.9 cm^2^/m^2^	PC (head of pancreas)	Curative	6★
Williet et al. (2021) [19]	France	Retrospective	Single-center	79	66 (58.5–74)	45.5	55	69.6	SMI	Optimum stratification (log rank, outcome: ‘mortality’)	M < 55 cm^2^/m^2^ F < 39 cm^2^/m^2^	Metastatic PDAC	Palliative	7★
Wu et al. (2019E) [14]	Taiwan	Retrospective	Single-center	146	65.5 (36.7–92.2)	63	16	10.9	TSM (=SMI)	Fujiwara et al.	M < 36.2 cm^2^/m^2^ F < 29.6 cm^2^/m^2^	PC	Not specified	8★
Wu et al. (2019W) [14]	Taiwan	Retrospective	Single-center	146	65.5 (36.7–92.2)	63	97	66.4	TSM (=SMI)	Prado et al.	M < 52.4 cm^2^/m^2^ F < 38.5 cm^2^/m^2^	PC	Not specified	8★
Zhang et al. (2023) [54]	China	Retrospective	Single-center	113	59 (33–84)	41.5	49	43.3	SMI	Zeng et al.	M < 44.77 cm^2^/m^2^ F < 32.50 cm^2^/m^2^	PC	Curative	8★

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
