# Peer review of "Prevalence of Sarcopenia Determined by Computed Tomography in Pancreatic Cancer: A Systematic Review and Meta-Analysis of Observational Studies"

_cancers, 2024, doi:10.3390/cancers16193356_

Round 1

Reviewer 1 Report

Comments and Suggestions for Authors

In the manuscript, the authors present a systematic review regarding the incidence of sarcopenia in patients with pancreatic cancer. The general idea of the manuscript is quite interesting. In my opinion, in order to improve the quality of the manuscript, some changes have to be done. My  observations are :

- please include some data regarding the stages of the disease and about the histology of the tumor. The survival rates of this cases depends on the stages of the disease and also on the histological type of the tumor. 

Author Response

REVIEWER 1:

In the manuscript, the authors present a systematic review regarding the incidence of sarcopenia in patients with pancreatic cancer. The general idea of the manuscript is quite interesting. In my opinion, in order to improve the quality of the manuscript, some changes have to be done. My  observations are :

- please include some data regarding the stages of the disease and about the histology of the tumor. The survival rates of this cases depends on the stages of the disease and also on the histological type of the tumor. 

Thank you very much for your positive suggestion. Unfortunately, many of the included studies do not provide detailed information on tumor histology or stage, which indeed precluded us from carrying out subgroup analyses for these variables. Please also note that the “tumor information” column of Table 1 includes the available information on both PC stage and histology as provided in the original studies.

As examples of the limited information available regarding these variables in the included studies, I reproduce the information provided in the original articles regarding these data in the first 3 papers included in Table 1, which is as follows:

-Amini et al.: “All patients who underwent curative intent resection for pancreatic adenocarcinoma at the Department of Surgery of the Johns Hopkins Hospital between April 1996 and March 2014 were identified”. à This was included in Table 1 (PDAC [=pancreatic ductal adenocarcinoma, as explained in the Table legend], curative intention)

-Basile et al.: “This is an observational, retrospective study that examined data of 165 advanced PC patients treated at the Oncology department of Udine (…). All patients had confirmed PC. [All patients received chemotherapy].

-Beetz et al.: “All patients were diagnosed with pancreatic adenocarcinoma (…). Patients with other types of pancreatic cancer (…) were excluded”. There is no specific information of patient management in the paper, so Table 1 states it as “not specified”.

However, considering your relevant suggestion, we have clarified the information provided and where to consult it at the end of section 3.1., and added this point as an important issue to be addressed in future studies in the last paragraph of the discussion (please note that in the previous paragraph we had already mentioned the need for further addressing tumor histology as a potentially relevant factor in this context).

Reviewer 2 Report

Comments and Suggestions for Authors

Very strong work, but I think could be interesting divided palliative treatment to curative treatment. Infact in advanced cancer sarcopenia is common. Otherwise the high variety of incidence could be explained in curative series if there is a surgical treatment or not.

Comments on the Quality of English Language

Very comprensive

Author Response

REVIEWER 2:

Very strong work, but I think could be interesting divided palliative treatment to curative treatment. Infact in advanced cancer sarcopenia is common. Otherwise the high variety of incidence could be explained in curative series if there is a surgical treatment or not.

Thank you for your positive evaluation. We are not certain about the specific suggestion of the reviewer, since we already provided a subgroup analysis based on treatment intention (i.e., palliative vs curative setting), as shown in Figure 5 and in the specific subsection of section 3.1. Nonetheless, we have included some new lines in the discussion section regarding this point to further stress the importance of the oncological status of patients.

Reviewer 3 Report

Comments and Suggestions for Authors

The systematic review “Prevalence of sarcopenia determined by computed tomography in pancreatic cancer: a systematic review and meta-analysis of observational studies” by Láinez Ramos-Bossini and team aims to evaluate the prevalence of sarcopenia in pancreatic cancer patients using computed tomography, further exploring the impact of measurement criteria on the reported prevalence. It is interesting to see the outcome of the analysis they performed however it cannot be generalized at this stage. I agree that it demands standardized criteria in sarcopenia assessment to enhance comparability and accuracy in clinical practice and research. 

Before further considering this review for publication I have a few comments. 

  1. The style of the abstract is boring and there is no need to classify it as introduction, methods etc. The flow of the abstract is not appealing and I recommend rewriting it.

  2. Introduction doesn’t provide sufficient information. Please provide more background information.

  3. Figure captions can be more detailed. Always consider the readability so that one can understand the experiment/analysis simply by looking at the figure and figure legend.

  4. The authors state that “ These results should be also extrapolated to other oncological conditions and scenarios” and it is highly speculative without minimum background information. I would rather elaborate in a few sentences or as a separate section. Please provide more information.

  5. Results are not described with clarity. I would address this study for a more general audience considering the reach of the journal. Additional information and explanation is necessary throughout the main text of the results section. 

  6. Although not relevant at this stage, I wonder why the sections of author contribution, funding and COI are empty?

The review needs significant improvement to be considered for publication. I encourage the authors to revise it addressing a general audience.

Author Response

REVIEWER 3:

The systematic review “Prevalence of sarcopenia determined by computed tomography in pancreatic cancer: a systematic review and meta-analysis of observational studies” by Láinez Ramos-Bossini and team aims to evaluate the prevalence of sarcopenia in pancreatic cancer patients using computed tomography, further exploring the impact of measurement criteria on the reported prevalence. It is interesting to see the outcome of the analysis they performed however it cannot be generalized at this stage. I agree that it demands standardized criteria in sarcopenia assessment to enhance comparability and accuracy in clinical practice and research. 

Before further considering this review for publication I have a few comments. 

  1. The style of the abstract is boring and there is no need to classify it as introduction, methods etc. The flow of the abstract is not appealing and I recommend rewriting it.

Thank you for your comment, with which we agree. We have rewritten the abstract to make it more enticing according to your suggestion. We preserved the relevant information that needs to be included according to the PRISMA guidelines, but we think that the revised version is quite clearer, direct, and informative.

2. Introduction doesn’t provide sufficient information. Please provide more background information.

We have included new content to provide further introductory information for the paper. In our opinion, the essential ideas for the reader of the paper are well exposed and the rationale of the paper is clear, including the reason why sarcopenia is relevant in pancreatic cancer, the role of CT as an easy-to-obtain method for its measurement, the existing variability for its cutoff using different proposed methods and how it may influence clinical outcomes, including the most relevant paper that directly addressed this issue.

3. Figure captions can be more detailed. Always consider the readability so that one can understand the experiment/analysis simply by looking at the figure and figure legend.

We have now provided a more detailed explanation to each of the figure legends. In addition, according to another reviewer’s suggestion, we have included the references to each of the included studies.

4. The authors state that “ These results should be also extrapolated to other oncological conditions and scenarios” and it is highly speculative without minimum background information. I would rather elaborate in a few sentences or as a separate section. Please provide more information.

Thank you, we agree that the way it was expressed is too vague. The underlying idea is that the need for standardizing how to measure CT-based sarcopenia is not exclusive to pancreatic cancer, but to cancer in general. This is quite obvious, but still needs to be stated since it emphasizes a general limitation on this increasingly enticing research field, especially given the broad interest of the journal in relation to cancer (not just pancreatic cancer). We have modified the sentence (rewording “extrapolate” to “explore”), so as to open the door to further research on the topic. We think this way the sentence is clear and the conclusions more refined.

5. Results are not described with clarity. I would address this study for a more general audience considering the reach of the journal. Additional information and explanation is necessary throughout the main text of the results section. 

We have carefully revised the Results section, but your comment is not sufficiently clear to us. We have previous experience in meta-analysis publications and we think that results need to be aseptic and as self-explainable as possible, and we tried to follow this philosophy. We have come to the conclusion that the results may be more understandable by explicitly creating subsections for each subgroup analysis under section 3.2. We also made some minor clarifications throughout the results section to facilitate readiness; for instance, we simplified the reported percentages in the text .Please note that to better understand the subgroup analyses, information provided in the 2.3. subsection of the Methods is important. We hope that the changes made have improved the clarity of the manuscript.

6. Although not relevant at this stage, I wonder why the sections of author contribution, funding and COI are empty?

We have now provided the corresponding information for these sections..

The review needs significant improvement to be considered for publication. I encourage the authors to revise it addressing a general audience.

Thank you. We hope that the changes made to the manuscript meet the reviewer’s standards now.

Reviewer 4 Report

Comments and Suggestions for Authors

The wide variation in the incidence of sarcopenia reported in the study  has different motivations, including those of a racial and socio-economic nature. However, preoperative evaluation of sarcopenia is important to define the extent of the surgical risk and the onset of complications.The use of Radiomics may represent a valid aid. The article may be published if the authors mention these issues. 

The Authors  should make a mention of these issues

Author Response

REVIEWER 4:

The wide variation in the incidence of sarcopenia reported in the study  has different motivations, including those of a racial and socio-economic nature. However, preoperative evaluation of sarcopenia is important to define the extent of the surgical risk and the onset of complications.The use of Radiomics may represent a valid aid. The article may be published if the authors mention these issues. 

The Authors  should make a mention of these issues

Thank you for these suggestions, with which we agree and which have now been included to the introduction and discussion sections. We hope they are sufficient so as to make the readers aware of the importance of addressing the said (and other) limitations, and to pay attention to Radiomics as a potential aid in the future.

Round 2

Reviewer 3 Report

Comments and Suggestions for Authors

I appreciate the effort of the authors to incorporate my suggestions. By including the suggested modifications and explanation for other questions the authors have addressed most of my concerns and suggestions. This is now a good review and I fully support the publication of this manuscript in ‘cancers’.